# A DIFFUSION MODEL ON TORIC VARIETIES WITH APPLICATION TO PROTEIN LOOP MODELING

## ABSTRACT

The conformation spaces of loop regions in proteins as well as closed kinematic linkages in robotics can be described by systems of polynomial equations, forming Toric varieties. These are real algebraic varieties, formulated as the zero sets of polynomial equations constraining the rotor angles in a linkage or macromolecular chain. These spaces are essentially stitched manifolds and contain singularities. Diffusion models have achieved spectacular success in applications in Cartesian space and smooth manifolds but have not been extended to varieties. Here we develop a diffusion model on the underlying variety by utilizing an appropriate Jacobian, whose loss of rank indicates singularities. This allows our method to explore the variety, without encountering singular or infeasible states. We demonstrated the approach on two important protein structure prediction problems: one is prediction of Major Histocompatibility Complex (MHC) peptide interactions, a critical part in the design of neoantigen vaccines, and the other is loop prediction for nanobodies, an important class of drugs. In both, we improve upon the state of the art open source AlphaFold.

## 1 INTRODUCTION

Proteins are essential polymeric biological molecules, and knowing the 3D structure of a protein is key for our ability to understand its function. A protein loop is a non-regular contiguous segment of the protein chain which connects the regular structural elements, such as alpha helices and beta sheets, shown in Fig. 1(a). The "non-regularity" of loop structure is expressed as a lack of a fixed periodic pattern of hydrogen bonding typical for alpha helices and beta sheets. This absence of stabilizing hydrogen bonding, compounded by the fact that loops are often located on the surface of the protein and thus exposed to the solvent, makes loop structures more challenging to characterize both experimentally (using X-ray crystallography or Cryo-Electron Microscopy) and computationally in the context of protein structure prediction [Barozet et al. (2021)].

At the same time, protein loops often play key roles in protein function, forming the components of enzymatic sites (such as kinase activation loops, HIV protease flap loops, Dihydrofolate Reductase Met20 loop, etc. [Malabanan et al. (2010)]) as well as serving as the binding sites for other molecules, such as most prominently in Complementarity Determining Regions (CDR) of antibodies [Nowak et al. (2016)]. This discrepancy between functional importance and our limited ability to model them computationally (compared to protein structure in general) makes the problem of predicting the protein loop structures highly relevant, and serves as a motivation for the work presented here.

Protein structure prediction poses a challenge to the scientific community. Recently, the progress has been accelerated by AlphaFold 2 (AF2) [Jumper et al. (2021)], made possible by the advances in the field of Machine Learning and availability of data resulting from the decades long accumulation of experimental structures deposited in the Protein Data Bank (PDB) [Berman et al. (2009)]. However, even such advanced approaches often have difficulties predicting certain structural elements, with loops being a prime example (see Fig. 1(b)). These limitations call for the development of novel computational methods.

The general problem of protein structure prediction can be formulated as a generative task of learning the probability distribution $p(x|s)$ over protein structures $x$ conditioned on protein sequence $s$ (in the case of partial modeling, like in case of protein loops, we additionally condition the distribution on the non-loop portion of the protein structure $r$ and deal with the target distribution $p(x|r, s)$).

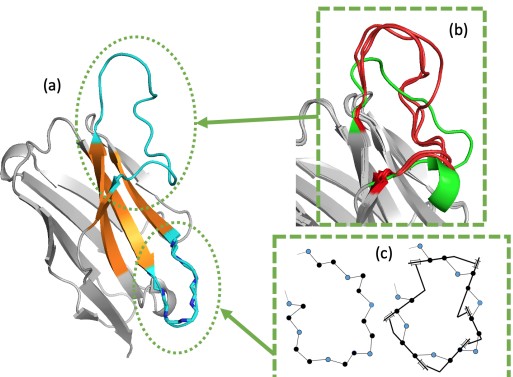

Figure 1: **a:** A structure prediction from AF2 (PDB ID: 8J5J). The cyan segments show two loop regions of the structure, which is connecting beta sheet segments in orange. **b:** The CDR3 loop region in the experimental structure is in green, the top 3 predictions from AF2 are in red while the rest of the structures are in gray. The predictions of CDR3 loop region are close to each other but not matched with the experimental structure, while other segments are nearly matched. **c:** The backbone of a loop region can be extracted and converted to a closed 6-revolute kinematic linkage.

The idea has received a lot of attention recently, with latent variable generative models, especially diffusion models, being successfully used to generate the structures of proteins and other molecules, including the modeling of protein structures [Watson et al. (2023)], protein backbones [Yim et al. (2023)], small molecules [Xu et al. (2022); Jing et al. (2022)], and in molecular docking [Corso et al. (2023)].

The earlier generation of such models does not explicitly incorporate the constraints imposed by chemical bonding and applies the noise to 3D coordinates of each atom independently when constructing the forward process, such as in [Xu et al. (2022); Hoogeboom et al. (2022); Yim et al. (2023); Watson et al. (2023)]. This practice is often referred to as diffusion in Euclidean space, and leads to an increased number of denoising steps as all features of the chemical structure have to be learned from data directly. More recently, a number of approaches have taken advantage of the fact that molecular flexibility is largely limited to the so-called torsional angles formed by a sequence of 4 atoms connected consecutively by three covalent bonds, while bond length and 3-atom bond angles maintain essentially constant values. Representing molecular structure in terms of torsional angles significantly reduces the dimensionality of the problem and has been recently used to construct more efficient diffusion models for small non-protein molecules [Jing et al. (2022); Corso et al. (2023)].

While the above approaches perform very well for tree-like molecular graphs, the case of protein loops adds additional geometric constraints to the picture, namely the requirement of loop closure: the generated loop structures must have both their ends fixed, while the protein chain must remain unbroken, i.e. all chemical bond lengths fall within acceptable margins of error from expected values. The closure condition makes the torsional spaces of closed loops highly constrained subspaces of the hypertori which may involve singularities, and are therefore challenging to learn directly, especially since every loop has a different submanifold from others, determined by its length and the relative position of the two ends. Incorporating the closure constraint into the model as an inductive bias reduces the effective dimensionality of the problem and may produce an architecture which is both more computing- and data-efficient (similarly to how torsional diffusion is more efficient than Euclidean) relative to both torsional diffusion and Euclidean diffusion baselines. Here we are proposing such a diffusion model operating on toric varieties which can be used to study the constrained manifold for the loop regions in proteins.

The main contributions of the work are:

1. We propose a diffusion-inspired method suitable for toric varieties and applicable to the problem of structure prediction for a broad class of constrained molecules, including protein loops, macrocycles [Jimenez et al. (2023)], stapled peptides [Li et al. (2020)] and MHC-bound peptides.

2. We demonstrate the performance of the architecture on two important biological problems: 1) predicting the structures of peptides bound to MHC receptors and 2) predicting the structures of nanobody CDR3 loops. On an MHC type I dataset containing 78 complexes, we achieve over 15% improvement in terms of median RMSD over the public domain state of the art model AlphaFold 2. Similarly, on a nanobody dataset containing 38 cases we achieve over 20% improvement.

## 2 BACKGROUND AND RELATED WORK

A conformation of a molecule can be represented by coordinates of all atoms in Euclidean space, which can be thought of as an element in $\mathbb{R}^{3N}$, where $N$ is the number of atoms in the molecule. However, the observed variations of bond lengths and angles in experiments are relatively small, and the flexibility of a molecule is mainly determined by the torsional angles at rotatable bonds [Gō & Scheraga (1970); Dinner (2000)]. In the case of proteins, we have chains of amino acids joined by peptide bonds. Each amino acid contributes three atoms to the protein backbone, a Nitrogen and two Carbons, so that a protein of $M$ amino acids entails the backbone of $3M$ atoms, $\{A_i(N_i - C_{\alpha,i} - C_i)\}_{i=1}^M$, and $3M - 2$ rotatable bonds. The peptide bonds $(C_{i-1} - N_i)$ formed by a dehydration reaction between two consecutive amino acids, $A_{i-1}$ and $A_i$, are usually treated as non-rotatable because they tend to have small changes in the structures. Thus, the embedding space of an internal protein backbone conformations is a hypertorus with dimension $2M - 2$. For the purpose of this work, we assume the internal conformation of side chains (short molecular chains branching from the $C_\alpha$ atom and specific for each type of amino acid) to be fixed (but note in passing that as side chain conformations are not subject to closure constraints, they can be straightforwardly handled by using existing approaches based on torsional diffusion, and our architecture can be augmented to include such treatment). In this case, the remaining flexibility in the protein chain comes from the $\phi$ and $\psi$ angles in the backbone (resp. torsions $(C_{i-1}, N_i, C_{\alpha,i}, C_i)$ and $(N_i, C_{\alpha,i}, C_i, N_{i+1})$). Here we focus on the movement of the backbone of a protein loop region which is constrained by its attachment at both ends to the rest of the structure. The backbone of a loop and a schematic of its conversion to a linkage is shown in Fig. 1(c).

To the best of our knowledge, there is no deep learning method to generate loop conformers in toric variety space, but several methods have been proposed to explore the conformational space of loops, such as systematic sampling, Molecular Dynamics (MD), Monte Carlo (MC), and geometric methods [Barozet et al. (2021)]. In MD and MC methods, an ensemble of conformations is obtained through computationally intensive simulations. In systematic sampling, with rigid rotor assumption where only the torsional angles are flexible [Gō & Scheraga (1970)], different conformations of the loop can be explored through sampling the backbone torsional angles $\phi, \psi$ with a given granularity. This method is exhaustive and deterministic, but the optimal granularity varies for different molecules. Without considering the constraints at two ends, the generated conformers are usually open. As the structures of molecules can be treated as geometric objects, some geometric methods were also proposed to sample the loop regions, such as Triaxial Loop Closure [Coutsias et al. (2004)] and constrained normal mode analysis (NMA) [López-Blanco et al. (2022)]. In [Coutsias et al. (2004)], the kinematic view of the loop was explored. The fully algebraic method can explicitly account for the closure constraints imposed by having the two ends of the loop fixed. This method was then extended to the KIC method in the Rosetta suite for molecular modeling [Mandell et al. (2009); Stein & Kortemme (2013)]. In [López-Blanco et al. (2022)], constraints of loop closure were added to regular NMA method to explore the local conformation of loops. However, in such geometric methods, thousands of conformations need to be generated first and the representative conformers can be chosen through clustering based on pairwise root mean square deviation (RMSD). In geometric methods, plenty of redundant conformations will be generated and an energy-based scoring function is still needed to rank the conformations.

Finally, it should be acknowledged that the state of the art in protein structure modeling is currently best exemplified by the AF2 approach [Jumper et al. (2021)] (and the recently made available AlphaFold 3 [Abramson et al. (2024)]), which represents a major breakthrough in the general purpose structural modeling of proteins and sets a new standard for prediction accuracy. Given the prediction accuracy from AF2, in most realistic modeling scenarios, the loop-specific modeling tools will use the predictions made by AF2 or RoseTTAFold [Baek et al. (2021)] as a starting point, refining the predictions of the loop regions while keeping the rest of the structure largely intact. In this context,

any improvements that loop-specific modeling tools aim to achieve have to be characterized relative to the predictions of the baseline general-purpose model.

## 3 METHOD: DIFFUSION ON TORIC VARIETIES

### 3.1 OVERVIEW

Unlike a free chain, the backbone angles in an $n$-torsion loop are subjected to additional constraints that keep ends fixed with respect to the rest of the protein. These constraints define a toric algebraic variety on the hypertorus $T^n$, which is essentially stitched manifolds possibly featuring singularities. Mathematically, the object of interest is the $(n-6)$-dimensional subvariety of the $n$-Torus defined by a system of trigonometric expressions relating two ends of the loop through a sequence of orthogonal transformations defined by six pivotal rotors along the closed kinematic chain. Expressing all sines and cosines in terms of half-tangents of the six constrained torsions, these trigonometric closure conditions result in a system of polynomials whose real solutions define the alternative conformations of the loop (derivations of the polynomial system can be found e.g. in [Cao et al. (2023)], see also [Angeles (2014), p.375-389]). Standard methods reduce the problem to the solution of a 16-degree polynomial in one of the variables, and from each real root the remaining five variables are determined. The real zero set of the closure polynomial system can have nontrivial topology. This introduces nontrivial variety structure, e.g., the space of a closed canonical octagonal chain is topologically the union of a sphere and a Klein bottle that intersect along two circles [Martin et al. (2010)].

We approach the loop modeling problem by learning a probability distribution $p(\mathbf{x}|\mathbf{r}, \mathbf{s})$ over loop conformations $\mathbf{x}$ conditioned on the remaining protein structure $\mathbf{r}$ and sequence $\mathbf{s}$. For that, we develop a diffusion model operating on toric varieties. Unlike diffusion on Euclidean spaces or smooth manifolds, diffusion on varieties can be challenging in the vicinity of singularities. We propose a way relying on the tangent space to move as shown in Fig. 2, in the spirit of the Geodesic Random Walk [De Bortoli et al. (2022)]. At each step, the tangential noise is sampled and then the tangent vector is projected back to the variety through a map to produce a valid step on the variety. We achieve this by applying the $R6B6$ [Cao et al. (2023)] algorithm to maintain loop closure. $R6B6$ (from "6 Rotors/6 Bars") is a robust algorithm to handle loop closure and conformational sampling problems in chains with fixed ends. It uses a system of polynomial equations to solve for the constrained torsions, ensuring the chain remains closed while allowing flexible perturbation of the remaining torsions. In a chain with $n$ flexible torsions, we can select $n-6$ torsions to perturb, and $R6B6$ can be used to solve for the remaining 6 torsions to maintain two ends of the chain fixed with respect to the rest of the protein structure. To add noise to the diffusion process we resort to the Jacobian matrix constructed from the geometrical loop closure relationships to obtain an orthogonal set of basis vectors for the space of infinitesimal deformations consistent with loop closure. The following three sections present the algorithmic details of our method.

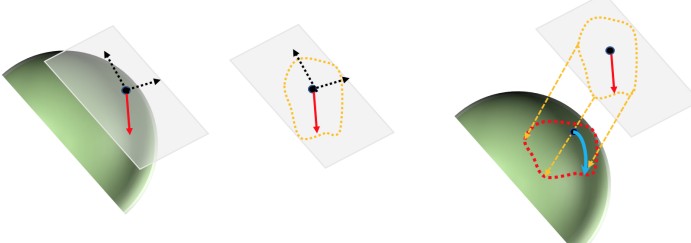

Figure 2: The green denotes the variety and the white plane is the tangent space at the point. The tangential noise is sampled (red line) based on the basis vectors (dashed black lines) of the tangent space. The tangent vector is then projected back (orange dashed lines with arrows) to produce a geodesics step on the variety (blue). The orange curved region denotes the boundary of the movement in the tangent space and the red curved region is the boundary of the movement at current step on the variety.

## 3.2 DIRECTIONS OF CONCERTED MOVEMENT FOR A LOOP REGION

Consider a loop $\{\mathbf{R}_i\}_{i=1}^n$ with $n \geq 6$ flexible backbone torsions $\zeta_i, i = 1, ..., n$, the ends of which are fixed. The torsions $\zeta_i$ are all flexible but should be chosen properly to construct realizations of the loop that are consistent with the closure constraints. Under certain conditions, we can ensure loop closure by assigning the values for $n - 6$ torsions and solving a system of polynomial equations to determine the remaining 6 [Angeles (2014)]. Thus the torsional space of the loop is a $(n - 6)$-dimensional variety embedded in an $n$-dimensional torus, and the dimension of the tangent space at a regular point is also $n - 6$. To characterize the tangent space, we consider a concerted change of all torsions $\zeta \to \zeta + d\zeta$ in the loop that keeps its ends fixed. At any point $\mathbf{R}$ of the chain past the fixed ends, we should have that

$$0 = d\mathbf{R} = \sum_{i=1}^n \mathbf{\Gamma}_i \times (\mathbf{R} - \mathbf{R}_i)d\zeta_i \Rightarrow \left(\sum_{i=1}^n \mathbf{\Gamma}_i d\zeta_i\right) \times \mathbf{R} - \left(\sum_{i=1}^n \mathbf{\Gamma}_i \times \mathbf{R}_i d\zeta_i\right) = 0, \quad (1)$$

where $\mathbf{\Gamma}_i$ is the unit vector along the $i$th torsional rotation axis and $\mathbf{R}_i$ is the position of $i$th atom. Since this is true for arbitrary $\mathbf{R}$, both expressions in parentheses of Equation 1 must vanish independently, from which we find

$$\mathbf{P}d\zeta = \sum_{i=1}^n \mathbf{P}_i d\zeta_i = 0, \mathbf{P} := (\mathbf{P}_1 \ \mathbf{P}_2 \cdots \mathbf{P}_N) \text{ where } \mathbf{P}_i = \begin{pmatrix} \mathbf{\Gamma}_i \\ \mathbf{\Gamma}_i \times \mathbf{R}_i \end{pmatrix}, \quad (2)$$

where $\mathbf{P}$ is the Jacobian matrix whose dimension is $6 \times n$. The columns of the Jacobian are the Plücker coordinates [Angeles (2014), p.102] of the corresponding axes. Basic analysis (the Implicit Function Theorem) guarantees that six of the variables may be expressed as differentiable functions of the remaining ones provided $\mathbf{P}$ has full rank. In that case there exist at least 6 independent columns, so that the corresponding 6 torsional perturbations can be expressed locally as differentiable functions of the other $n - 6$ torsions. Intuitively, the linkage needs 6 DoF to maintain closure, since given the location of one end, placing the other at the correct position and orientation requires, roughly speaking, 3 translational and 3 rotational degrees of freedom. From the singular value decomposition (SVD) of the $6 \times n$ matrix $\mathbf{P}$, we can obtain a set of $n - 6$ orthonormal null vectors $\mathbf{v}_i, i = 1, ..., n - 6$ with $\mathbf{v}_i \cdot \mathbf{v}_j = 0$ and $||\mathbf{v}_i|| = 1$, forming the basis for the tangent space of the variety at current point. These vectors in the tangent space provide a set of orthogonal directions for the concerted movement of torsional angles in the loop. Any infinitesimal perturbation of the loop torsions that can be expressed as a linear combination of the vectors $\mathbf{v}_i$ in the tangent space, will keep the loop closed at both ends.

## 3.3 TRAINING AND INFERENCE

We propose a denoising diffusion model to approximate the distribution over loop torsions conditioned on protein sequence and structure. We train a score model $\mathbf{s}_\theta(\boldsymbol{x}_t, t)$ in the tangent space $span(\{\mathbf{v}_i, i = 1, .., n - 6\})$ of the closure variety, where $\boldsymbol{x}_t$ is the state of geometric graph describing the protein structure at time $t$. The score model therefore predicts a vector $\delta\boldsymbol{\tau}$ living in $span(\{\mathbf{v}_i, i = 1, .., n - 6\})$ that can be expressed in both ambient $n$-dimensional torsional basis and tangential basis $\{\mathbf{v}_i, i = 1, .., n - 6\}$ and is trained to match the score $\nabla_{\boldsymbol{\tau}_t} \log p(\boldsymbol{\tau}_t | \boldsymbol{\tau}_0)$ expressed in tangential basis, where $p(\boldsymbol{\tau}_t | \boldsymbol{\tau}_0)$ is the perturbation kernel of the forward diffusion.

To train the score model, we sample from $p(\boldsymbol{\tau}_t | \boldsymbol{\tau}_0)$ and compute its score. We chose the normal distribution as the kernel for the perturbation samples. The noise scale function is $\sigma_t = \sigma_{\min}^{1-t} \sigma_{\max}^t, t \in [0, 1]$. To add perturbation noise to the torsions of the loop, we first sample $(\tau_1, \tau_2, ..., \tau_{n-6})$ from $p(\boldsymbol{\tau}_t | \boldsymbol{\tau}_0)$, components of the perturbation in tangential basis. The resulting perturbation to the $n$ torsions is given by:

$$\Delta\boldsymbol{\zeta}_t = \tau_1 \mathbf{v}_1 + \tau_2 \mathbf{v}_2 + .. + \tau_{n-6} \mathbf{v}_{n-6}.$$

However, this tangential perturbation may push us off the variety, breaking the loop. To maintain closure, $R6B6$ algorithm is applied to project the movement back to the variety (as shown in Fig. 2). We select 6 largest components in $\Delta\boldsymbol{\zeta}_t$ and set the corresponding torsions as unknowns, after verifying that the corresponding $6 \times 6$ submatrix of the Jacobian is invertible. We next add the remaining $n - 6$ components for the corresponding $n - 6$ torsions. The solutions for the 6 selected torsions can be solved by $R6B6$, i.e. guarantee that the spatial constraints at two ends of the loop are exactly satisfied. The difference $\Delta\boldsymbol{\zeta}_t'$ between the obtained torsion values and the original values before perturbation

provides the noise to these 6 torsions. In the training, if the closure problem is not solvable by $R6B6$ which indicates an infeasible movement, the next perturbation will be tried until one solvable case is sampled successfully. In our experiments, the closure problem can almost always be solved after one perturbation, with rare failures requiring at most three trials. During training, we sample the time $t$ uniformly and minimize the loss $\mathcal{L}(\theta) = \mathbb{E}_t[\lambda(t)\mathbb{E}[||\mathbf{s}_\theta(\boldsymbol{\tau}_t, t) - \nabla_{\boldsymbol{\tau}_t} \log p(\boldsymbol{\tau}_t|\boldsymbol{\tau}_0)||^2]]$, where $\lambda(t) = \mathbb{E}[||\nabla_{\boldsymbol{\tau}_t} \log p(\boldsymbol{\tau}_t|\boldsymbol{\tau}_0)||^2]$ as in [Song et al. (2021); Corso et al. (2023)]. The procedures for training are given in Algorithm 1.

During the inference, the null vectors at current state will be computed by SVD and the tangential torsional movement $\delta\boldsymbol{\tau}$ can be predicted from the neural network $\mathbf{s}_\theta(\boldsymbol{x}_t, t)$, from which we can obtain the proposed perturbation $\Delta\boldsymbol{\zeta}_t$. The algorithm $R6B6$ is used to check whether the loop is closed after perturbation, i.e. the moved point can be projected back to the variety. If the loop remains closed, the perturbation $\Delta\boldsymbol{\zeta}_t'$ will be added to the flexible torsions. Otherwise, the structure will stay at current state. The success rate of steps in the inference of our model is greater than 95%, and it takes approximately 1 second to produce one conformation with 20 denoising steps.

Each denoising step requires the usage of $R6B6$ and SVD. The computational cost of $R6B6$ is around 0.5 ms, while SVD for the $6 \times N$ Jacobian matrix is $O(6N \min(6, N))$, which is linear in $N$. Since $N$ is at most 34 in our use case, this results in a computation time on the order of $10^{-5}$ seconds per step. Given that one diffusion step requires less than 0.1 seconds overall, the cost of SVD is negligible compared to the benefits it provides in efficiently sampling the variety. With all the flexible torsions in the loop updated, we can reconstruct the structure accordingly, which has two ends of the loop closed. The procedures for inference are given in Algorithm 2.

---

**Algorithm 1** Training

---

    **Input** Molecular graphs $[G_0, G_1, ..., G_N]$, learning rate $\alpha$
    **Output** Score model $\boldsymbol{s}_\theta$
    **for** $epoch = 1$ to $epoch_{\max}$ **do**
        **for** $G$ in $[G_0, G_1, ..., G_N]$ **do**
            extract loop region $\mathbf{l}_p$;
            compute all null vectors $\mathbf{v}_i$ using Jacobian based on $\mathbf{l}_p$;
            sample $t \in U[0, 1]$;
            set close flag $flag = 0$;
            **while** $flag = 0$ **do**
                sample $\Delta\boldsymbol{\tau}$ from Gaussian $p_{t|0}(\cdot|0)$ with $\sigma_t = \sigma_{\min}^{1-t}\sigma_{\max}^t$;
                $\Delta\boldsymbol{\zeta}_t = \sum \Delta\boldsymbol{\tau}_i \cdot \mathbf{v}_i$;
                $\Delta\boldsymbol{\zeta}_t' = Closure(\Delta\boldsymbol{\zeta}_t)^*$
                If $\Delta\boldsymbol{\zeta}_t'$ is not None: $flag = 1$;
            apply $\Delta\boldsymbol{\zeta}_t'$ to G;
            predict $\delta\boldsymbol{\tau} = \boldsymbol{s}_{\theta,G}(t)^{**}$;
            update $\theta \longleftarrow \theta - \alpha\nabla_\theta||\delta\boldsymbol{\tau} - \nabla_{\Delta\boldsymbol{\tau}}p_{t|0}(\Delta\boldsymbol{\tau}|0)||^2$;
    $^*Closure(\Delta\boldsymbol{\zeta}_t)$: 1. select 6 indices with largest components in $\Delta\boldsymbol{\zeta}_t$ as the pivots; 2. apply remaining $n - 6$ of $\Delta\boldsymbol{\zeta}_t$ to $\mathbf{l}_p$; 3. check if the loop can be closed by using $R6B6$: if true, modify $\Delta\boldsymbol{\zeta}_t$ to $\Delta\boldsymbol{\zeta}_t'$ and return $\Delta\boldsymbol{\zeta}_t'$, otherwise $\Delta\boldsymbol{\zeta}_t' = $ None.
    $^{**}\boldsymbol{s}_{\theta,G}(\mathbf{v}_i, t)$ first predicts scalars corresponding to the torsions followed by multiplication with all null vectors $\mathbf{v}_i$.

---

## 3.4 Architecture of Diffusion Model

We designed the score model $\boldsymbol{s}(\boldsymbol{x}, \boldsymbol{r}, t)$ to take as input protein structure represented as heterogeneous geometric graph in $3D$ including all atoms $\boldsymbol{x}$ in the loop region, and a coarse-grained $C_\alpha$ atom representation $\boldsymbol{r}$ of the residues for the remaining fixed part of the protein. Non-loop regions are fixed in the input graph, serve as spatial constraints to guide feasible loop movements while ensuring closure. $C_\alpha$ residue and all loop atom nodes are featurized with one hot amino acid type encoding. All nodes are sparsely connected based on distance cutoffs that depend on the types of nodes being linked and on the diffusion time. To account for roto-translation symmetries inherent to protein structure prediction problem, we use an architecture similar to SE(3)-equivariant Tensor Field

---

**Algorithm 2** Inference

**Input** Molecular graph $G'$, number of conformers $K$, number of steps $N$
**Output** Predicted ensemble $[G'_1, ..., G'_K]$
**for** $i = 1$ to $K$ **do**
    **for** $n = N$ to 1 **do**
        set $t = n/N$, $g(t) = \sigma_{\min}^{1-t} \sigma_{\max}^{t} \sqrt{2 \ln (\sigma_{\max}/\sigma_{\min})}$;
        extract loop region $\mathbf{l}_p$;
        compute all null vectors $\mathbf{v}_i$ using Jacobian based on $\mathbf{l}_p$;
        predict $\delta\boldsymbol{\tau} = \boldsymbol{s}_{\theta,G}(t)$
        draw $\boldsymbol{z}$ from Gaussian with $\sigma^2 = 1/N$;
        $\Delta\boldsymbol{\tau} = (g^2(t)/N)\delta\boldsymbol{\tau} + g(t)\boldsymbol{z}$;
        $\Delta\boldsymbol{\zeta}_t = \sum \Delta\boldsymbol{\tau}_i \cdot \mathbf{v}_i$;
        $\Delta\boldsymbol{\zeta}'_t = Closure(\Delta\boldsymbol{\zeta}_t)$;
        If $\Delta\boldsymbol{\zeta}'_t$ is not None, applying $\Delta\boldsymbol{\zeta}'_t$ to $\mathbf{l}_p$; otherwise skip this step;

---

Network [Thomas et al. (2018); Geiger & Smidt (2022)] which operates on the molecular geometric graph for interaction layers. The loop atom representations after the final interaction layer are subject to pseudotorque convolution at each rotatable bond. These convolutions produce roto-translation invariant torsional scores for all $n$ rotatable bonds in the loop. The vector of these scores is an $n$-dimensional vector in tangent space $T_\theta SO(2)^n$ of hypertorus. Our architecture is similar up to this point to that of [Corso et al. (2023)]. To account for closure condition, we project the torsional vector onto the tangent space $span(\{\mathbf{v}_i, i = 1, .., n-6\})$ of the closure subvariety at the current point. The basis vectors of this tangent space are obtained through SVD of the Jacobian matrix $\mathbf{P}$ introduced in Equation 2. The resulting $n$-dimensional projected vector is treated as the predicted score $\Delta\boldsymbol{\tau}$ and can be also expressed with $n-6$ coordinates in the $\{\mathbf{v}_i, i = 1, .., n-6\}$ basis. More details of the architecture can be found in Appendix B.

While our architecture is similar to SE(3)-equivariant Tensor Field Networks due to their robust handling of geometric symmetries, alternative architectures, such as PointNet [Qi et al. (2017)], could also be considered for processing point cloud data.

## 4  EXPERIMENTS

We evaluated our method on two important problems in protein structure prediction involving loop-like elements: predicting the structures of peptides bound to the Major Histocompatibility Complex and predicting the structures of nanobody complementarity-determining region loop 3 (CDR3). The structures were collected from PDB and the SAbDab [Schneider et al. (2022)], respectively, excluding any structures with missing loops. We used release time-based criteria to split the dataset; details are provided in Appendix C.1. In our diffusion model, a significant hyperparameter is the maximum noise level $\sigma_{\max}$. We set $\sigma_{\min} = \pi/100$ and examined $\sigma_{\max} = \pi/30, \pi/22, \pi/18, \pi/15, \pi/12, \pi/10$. Details of the hyperparameters are provided in Appendix Table 1.

While the training was done on PDB structures alone, during validation and testing we start our prediction from AF2 models [Jumper et al. (2021)] to ensure our method is not biased by the information contained in the native structures. Specifically, we used AF2 version 2.3 as implemented in ColabFold [Mirdita et al. (2022)] to predict the protein structures from their sequences, and these structures served as inputs for our trained diffusion model. Each structure was split into two parts: the loop region and the remaining part of the protein. The diffusion model then generated conformations for the loop regions.

In the experiments, we use AF2 as a baseline to compare our method against. While the main comparison we perform is to the starting structures generated with AF2 (one per case) as described above, we generate additional structures with AF2 to provide ensemble-level comparison (which becomes relevant as we generate multiple structures with our approach). It should be mentioned that the outputs from AF2 are in principle a deterministic function of its inputs, and it has been shown that the inputs can be stochastically subsampled to obtain an arbitrary number of diverse outputs [Del Alamo et al. (2022)]. Additional samples for comparison are generated by relying on this mechanism.

We must point out that we did not learn a model to assess the confidence of the generated ensembles. For that purpose, we performed local refinement and scoring of our predictions using AF2 similar to [Ghani et al. (2021); Roney & Ovchinnikov (2022)]. Specifically, we used predicted Local Distance Difference Test (pLDDT) values [Jumper et al. (2021)] to rank the generated structures. To evaluate the results, we computed the backbone RMSD between the refined conformations and the ground-truth loop regions after aligning the protein structures. The RMSD is given in Å, which is a unit of length often used in the field of structural biology and equal to $10^{-8}$ cm.

## 4.1 MHC CLASS I

For the MHC class I dataset, the MHC bound linear peptide has its two ends nearly fixed, while the intermediate part is free to move similar to a protein loop region. The distribution of peptide lengths in our dataset is given in Appendix C.2. We prepared a subset of 789 structures involving the peptides of lengths 9 and 10, the most common lengths in the dataset, and initially trained (636), validated (77) and tested (76) the model on this subset. The trained model was then applied to 78 peptides (released in years 2023 and 2024) with diverse lengths ranging from 8 to 11 residues.

Our predictions started from 1 seed of the 1st multimer model of AF2. We then ran 20 trajectories of 20 denoising steps each and used the resulting 20 structures for AF2-based refinement and pLDDT scoring (1st multimer model). We compared our predictions with those of AF2 and AF3. For AF2, we evaluated the structure used to initialize the diffusion denoising trajectories, as well as the top pLDDT prediction among 20 differently seeded AF2 predictions (1st multimer model). For AlphaFold 3 (AF3), we evaluated the five models produced by AF3 server [Abramson et al. (2024)]. The results are summarized in Table 1. One example for the results is shown in Fig. 3.

Overall, the prediction of peptides was improved by using diffusion model denoising. When selecting the model with top pLDDT for the peptide out of 20, the median RMSD decreased by 15.8% from 0.95 Å to 0.80 Å, and the mean decreased as well. We also provide RMSD values for other scenarios (including AF3) for reference.

| Top confidence model | Mean Å | Median Å |
| --- | --- | --- |
| AF2 | 1.20 | 0.95 |
| AF3* | 1.20 | **0.80** |
| Diffusion | **1.14** | **0.80** |

| Best RMSD model | Mean Å | Median Å |
| --- | --- | --- |
| AF2 | 1.13 | 0.93 |
| AF3 | 0.93 | **0.64** |
| Diffusion | **0.90** | 0.74 |

Table 1: RMSD comparison for MHC dataset predictions. * in AF3, the confidence function is different from AF2 and Diffusion.

## 4.2 NANOBODY CDR3 LOOPS

We next examined the performance of our approach on a nanobody dataset. The main determinants of nanobody-antigen binding are the sequence and structure of the nanobody CDR3 loop. Since the CDR3 sequence is known *a priori* [Chothia & Lesk (1987)], the prediction of CDR3 loop structure becomes the main point of interest, but is significantly complicated by the fact that these loops can be relatively long and hence flexible.

As longer loops pose a more significant challenge and are of high interest, we specifically focused on this class of difficult-to-model systems. We prepared a dataset of PDB nanobody structures containing CDR3 loops that are 15 to 20 residues long (505 structures in total). The distribution of the lengths of the loops is given in Appendix C.2. The dataset was initially split into 403 training, 51 validation, and 51 test samples. We next found our dataset to be significantly biased, and therefore removed samples with redundant CDR3 loops (same length and identical sequence) from the validation and test sets. This left us with 38 structures released in the years 2023 and 2024 in the final test set. For each case,

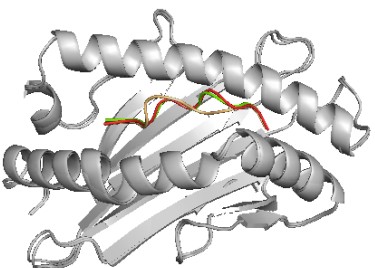

Figure 3: An example (PDB: 8ELG); the receptor proteins are shown in gray, the AF2 prediction (1.62 Å RMSD) is in orange, the PDB structure in red, and the top pLDDT Diffusion prediction (0.63 Å RMSD) in green.

our predictions started from the top pLDDT AF2 prediction (out of 5, all available AF2 monomer models with 1 seed each), while denoising (20 trajectories, 20 steps each), refinement, and scoring stages were the same as in the MHC case. Similarly to the MHC case, we compared our predictions with those of AF2 and AF3. For AF2, we only evaluated the structure used to initialize the diffusion denoising trajectories. For AF3, we evaluated the models produced by AF3 server. The results are summarized in Table 2. An example comparing AF2, PDB, and diffusion is shown in Fig. 4.

As can be seen in Table 2, the prediction of CDR3 loops was improved by using diffusion model denoising. Compared with the starting AF2 structure, our top pLDDT prediciton (out of 20) showed the median RMSD decreased by 22.5% from 2.00 Å to 1.55 Å, and the mean decreased by 14.3%. We also provide RMSD values for AF3 models for reference. It should be noted that confidence functions in AF2 and AF3 are different. Thus the top confidence models could not be compared directly unlike AF2 and Diffusion which use the same confidence function. As seen in the best RMSD rows of Table 2, the sampling of the diffusion model is comparable to AF3 which implies using AF3 confidence model can potentially further improve the top confidence model results.

| Top confidence model | Mean Å | Median Å |
|---|---|---|
| AF2 | 1.96 | 2.00 |
| AF3* | **1.37** | **1.19** |
| Diffusion | 1.68 | 1.55 |
| Best RMSD model | Mean Å | Median Å |
| AF2 | 1.73 | 1.67 |
| AF3 | **1.22** | 1.17 |
| Diffusion | 1.35 | **1.12** |

Table 2: RMSD comparison for nanobody dataset predictions. * in AF3, the confidence function is different from AF2 and Diffusion.

## 5 CONCLUSION

In this work, we presented a diffusion process on toric varieties, which can be applied to generate conformations for protein loop regions with constrained ends. We provided the first diffusion model to implement loop generation in torsional angle space. The performance of the method was demonstrated using the MHC dataset and the nanobody dataset. By generating and scoring a few conformations, the model's outputs improve upon the predictions from open source AlphaFold.

This model will benefit applications in protein design and drug discovery, as these fields often involve flexible loop regions and long-distance restraints in structures. Several extensions can be explored in

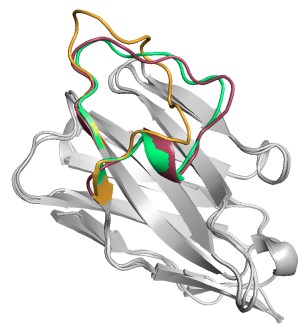

Figure 4: An example (PDB: 8J5J); the remaining parts of the proteins are shown in gray. The CDR3 loops from the PDB structure, AF2 prediction, and top pLDDT prediction from diffusion are displayed in red, orange (3.18 Å RMSD), and green (0.76 Å RMSD), respectively.

the future. A natural extension is to add flexibility to the rotatable bonds in amino acid side chains, which would provide a more complete description of structural movements. Moreover, diffusion on toric varieties could be applicable to other structurally constrained problems, such as macrocyclic molecule sampling and docking.

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
