## A Implementation Details

**Training.** We used the Adam optimizer Kingma & Ba (2014) for training the model. The exponential moving average of the weights during training and inference steps. We ran 20 denoising steps in the inference. Our final score-based diffusion model was trained on a single 48 GB RTX A6000 GPU and Intel Xeon CPU E5-1650 3.6GHz for 300 epochs (approximately 36 hours).

**Hyperparameters.** To determine the hyperparameters in the diffusion model, we trained smaller models with fewer than 0.5 million parameters before scaling up to the final model (5.5 million parameters). The smaller models were trained for 300 epochs. We used the percentage of improvement over the starting conformations with respect to the PDB structures to select the hyperparameters. The examined hyperparameters are listed in Table 1. The most important hyperparameter in tuning is the maximum noise level $\sigma_{\max}$. A large $\sigma_{\max}$ results in large movement during inference, which can break the constrained loop, while a small $\sigma_{\max}$ does not introduce noticeable changes in the structures. For the MHC dataset, the model trained with the hyperparameter $\sigma_{\max} = \pi/12$ produced the best results in the loop ensembles. We presented the results from the model trained with this parameter in the main text. For the nanobody dataset, the best results were obtained from the model trained with the hyperparameter $\sigma_{\max} = \pi/12$. The results from the model trained with this parameter are also shown in the main text.

| Parameter | Value |
|---|---|
| All atoms for remaining part of protein graph | NO |
| Use Language model embeddings | NO |
| Use hydrogens for ligands | NO |
| Use exponential moving average | YES |
| Maximum number of neighbors in protein graph | 24 |
| Maximum distance of the neighbors | 15 |
| Distance embedding method | Sinusoidal |
| Dropout | 0.1 |
| Learning Rate | 0.001 |
| Activation function | ReLU |
| Convolutional layers | 2 |
| Number of scalar features | 48 |
| Number of vector features | 10 |
| $\sigma_{\max}$ | $\pi/30, \pi/22, \pi/18, \pi/15, \boldsymbol{\pi/12}, \pi/10$ |

Table 1: Hyperparameter options for the score model. The parameter $\sigma_{\max}$, indicating the maximum noise level, was tuned for different datasets. For the MHC and nanobody datasets, the trained model with $\sigma_{\max} = \pi/12$ produced the best results in the loop ensembles.

## B Comparison between regular diffusion in torsional space and diffusion on toric varieties

In the diffusion on toric varieties, the basis vectors for the tangent space at a point on the manifold depend on the position of that point. Therefore, we have to compute the basis vectors at every step (both in the forward process and during denoising) using the Jacobian matrix $\mathbf{P}$ in Equation 2 and SVD. The differences between DiffDock Corso et al. (2023) (focusing on the torsional score part) and diffusion on toric varieties for the structure with $n$ flexible torsions are summarized in Table 2. In the DiffDock framework, the heterograph contains both the receptor and the ligand, and the neural network predicts the binding poses of the small molecule to the protein by modeling translation, rotation, and torsional changes in the ligand. The score model consists of several layers, including embedding layers, interaction layers, and a pseudoscalar layer. Diffusion on toric varieties is developed in a similar way. The graphs are constructed using detailed atomic representation for the loop and a coarse-grained representation for the rest of the protein. Moreover, the only flexibility in the loop arises from the backbone torsions $\phi$ and $\psi$, and translation and rotation of the loop are not considered. A comparison of the neural network structures for diffusion on toric varieties and DiffDock is shown in Fig. 1.

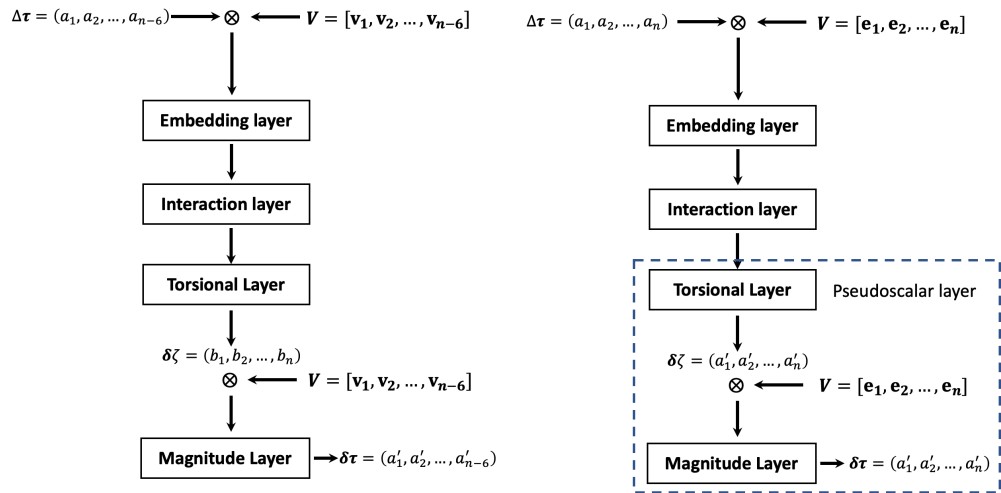

Figure 1: Schematic processes of training for diffusion on toric varieties (left) and DiffDock (right). The sampled noise $\Delta\tau$ is multiplied by the basis vectors $\mathbf{V}$ and then applied to the structure. The structure is next input to the embedding layer, interaction layer, and torsional layer in sequence. The torsional layer outputs scalars $\delta\zeta$ for the corresponding $n$ torsions. These scalars are finally multiplied with the basis vectors $\mathbf{V}$ in the magnitude layer to predict the magnitudes $\delta\tau$ in each direction of the basis vectors. In DiffDock, the combination of the torsional layer and magnitude layer is called the pseudoscalar layer because the standard basis vectors $[\mathbf{e}_1, \mathbf{e}_2, ..., \mathbf{e}_n]$ can be chosen for $\mathbf{V}$ and $\delta\tau = \delta\zeta$.

| | Toric varieties | Torsional space |
|---|---|---|
| Degrees of Freedom | $n - 6$ | $n$ |
| Basis vectors of tangent space | Null vector of Jacobian matrix | Standard basis of $\mathbb{R}^n$ |
| Projection function | $R6B6$ | Exponential map |

Table 2: Comparison between toric varieties diffusion and diffusion in torsional space for the structure with $n$ flexible torsions in the chain.

## C    DATA RELATED

### C.1    DETAILS OF DATA SPLIT

We split the MHC class I dataset into three parts: training, validation, and testing. The training dataset, consisting of 636 structures released up to September 30, 2020, and the validation set with 77 structures, released up to February 2, 2022, were used for training the model and optimizing hyperparameters. The test set with 76 structures, containing data released up to August 23, 2023, served as the first evaluation of the performance of the trained model. Similarly, we split the nanobody dataset into training (403 structures, released up to August 31, 2022), validation (51 structures, released up to August 2, 2023), and testing (51 structures, released up to May 1, 2024).

### C.2    DISTRIBUTION OF THE LENGTHS OF LOOPS

The distribution of the lengths of loops in MHC peptides and nanobody CDR loops are shown below.

## REFERENCES

Gabriele Corso, Hannes Stärk, Bowen Jing, Regina Barzilay, and Tommi Jaakkola. Diffdock: Diffusion steps, twists, and turns for molecular docking. In *International Conference on Learning Representations*, 2023.

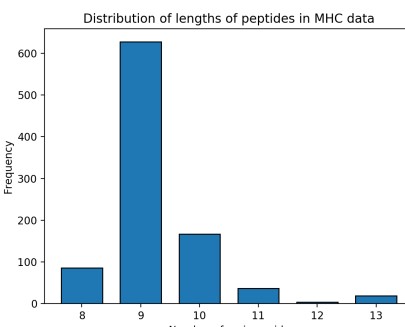 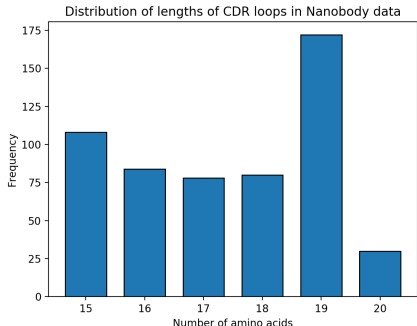

Figure 2: Length distributions for MHC peptides (Left) and nanobody loops (Right).

Diederik P. Kingma and Jimmy Ba. Adam: A method for stochastic optimization. *arXiv preprint arXiv:1412.6980*, 2014.