# OpenReview forum: "A diffusion model on toric varieties with application to protein loop modeling"
_ICLR.cc/2025/Conference — Submitted to ICLR 2025_

### Official Review · Reviewer_p9bC · 2024-11-03

**Soundness:** 1
**Presentation:** 3
**Contribution:** 2
**Rating:** 5
**Confidence:** 4

**Summary:**

The paper describes heuristic studies of random walks on spaces of chained atoms.

**Strengths:**

Two algorithms are given with pseudocodes.

**Weaknesses:**

Though the paper studies protein chains, they are called "molecules". In structural biology, a protein molecule can be a collection of many chains, which is called a bilogical assembly or a protein complex, while all smaller molecules (interacting with proteins) are called ligands.

Line 118: "we assume the internal conformation of side chains to be fixed"

This assumption seems too strict because side chains provide hydrogen bonds that form the most important secondary structures.

Line 120: "the space of internal protein chain conformations is the hypertorus."

Some values of torsion angles lead to clashed atoms involving side chains, hence a realizable space of torsion angles is smaller, see https://www.pnas.org/doi/abs/10.1073/pnas.1014674107

Lines 126-127: "Unlike diffusion on Euclidean spaces or smooth manifolds,
diffusion on varieties can be challenging in the vicinity of singularities."

Though singularities are explicitly mentioned here, Figure 1 and all further constructions discuss smooth manifolds. The word "variety" appears only two more times in the abstract.

Line 131: "a loop region can move in a finite number of directions to keep the ends fixed"

Because atom coordinates are continuous, they can move in infinitely many directions.

Lines 215: "The score model is using a similar architecture based on SE(3)-equivariant convolutional networks over point clouds"

If points are not ordered in these clouds, this case is not relevant for protein chains whose atoms are naturally ordered along the backbone. The ordered case is much easier than unordered because a complete SE(3)-invariant has a linear size.

**Questions:**

The major weakness is the lack of a rigorous problem statement.

Line 17: "we develop a diffusion model that performs a random walk"

What the purpose of this "random walk"?

The key concept of a "loop", which appears 80+ times in the paper including the title is never formally defined. Did the authors mean a geometric embedding of several consecutive amino acid residues or only some atoms (which?) along the backbone without side chains, which are not marked as alpha-helices or beta-strands?

Lines 114-115: "the observed variations of bond lengths and angles in experiments are relatively small"

How can this claim be justified? For protein chains, bond lengths and angles are not "observed" but are often fixed or refined to "ideal" values. Despite this artificial fixing, the Protein Data Bank has a large percentage of chains whose bond lengths and angles are outside many standard deviations from "ideal" values.

Lines 176-177: "(the Implicit Function Theorem) guarantees that six of the variables may be expressed as differentiable functions"

This theorem holds only under strict conditions. Why are they satisfied?

Lines 198-202: "We apply R6B6 Cao et al. (2023) algorithm, which is designed to solve closure problem in the chain with fixed ends ... if the closure problem is not solvable by R6B6 which indicating an infeasible movement, the next perturbation will be tried until one solvable case is sampled successfully."

Can these tries in a continuous space continue infinitely long?

Lines 285-287: "in the diffusion on toric verieties, the basis vectors for the tangent space at one point on the manifold are dependent on the position of the point, so we need to compute the basis vectors at every step"

There is no need to consider any tangent spaces of high-dimensional tori because these manifolds are flat (locally Euclidean in terms of distances and angles), see https://en.wikipedia.org/wiki/Flat_manifold. For a simple example, a 2D torus has two angle coordinates in ranges [0,2pi). Why do you need to embed this torus in 3D?

Line 301: "Each structure was split into two parts: the loop region and the remaining part of the protein."

While a loop region wasn't formally defined, a protein structure likely contains many loop regions. Which one is chosen here?

Line 308: "we used the same AF2 model to obtain predicted Local Distance Difference Test (pLDDT) values Mariani et al. (2013)"

Do you agree with the conclusion of Mariani et al. (2013) on page 2728: ``One disadvantage of the LDDT score is that it does not fulfill the mathematical criteria to be a metric"?

If yes, do you know that if a distance fails the triangle axiom with any positive error, then results of clustering can be pre-determined and hence not trustworthy as proved in https://ieeexplore.ieee.org/abstract/document/10574843?

Line 310: "We then used these values to rank the structures and determine the confidence of the predictions."

How are these predictions justified in the light of the references above?

Line 320-321: "The testing results showed that closer conformations to the PDB structures exist within the ensemble, demonstrating that the model functions correctly."

The loop regions have huge uncertainty in PDB structures because the structure determination methods usually have low resolutions in these regions. Is it possible justify that the "model functions correctly" by using unreliable "ground truth"?

How large are the databases discussed in sections 4.1 and 4.2?

Lines 13-14: "Toric varieties. These are real algebraic varieties, formulated as the zero sets of polynomial equations constraining the rotor angles in a linkage or macromolecular chain."

The abstract promised to describe toric varieties of macromolecular chains by polynomial equations. In what line numbers did these equations appear if the word "polynomial" was found twice in the abstract and never again?

**Details Of Ethics Concerns:**

The paper included the acknowledgments that de-anonymized the submission:

"This work was partly supported by NIH Grant RM1 GM135136, and by NSF Grant DMS 2054251. We gratefully acknowledge support from Laufer Center for Physical and Quantitative Biology. This research used resources of the Oak Ridge Leadership Computing Facility, which is a DOE Office of Science User Facility supported under Contract DE-AC05-00OR22725."

---

> ### Author Response · Authors · 2024-11-28
> **Response to Reviewer p9bC (part 1)**
>
> **General Comment**
>
> We thank the reviewer for their thoughtful feedback and careful review of our manuscript. We have revised the manuscript to improve its clarity and address the specific concerns raised.
>
> ---
>
> ### Responses to Concerns Raised in Section *Weaknesses*
>
>
> **1. The paper studies protein chains, but they are called "molecules." In structural biology, a protein molecule can be a collection of many chains, called a biological assembly or a protein complex, while all smaller molecules interacting with proteins are called ligands.**
>
> Thank you for pointing this out. We have adjusted the text throughout the manuscript to avoid potential confusion and ensure that the terminology aligns with conventions in structural biology. We would like to note, however, that our method is generalizable and can also be applied to other classes of constrained molecules, such as macrocycles.
>
> ---
>
> **2. Line 118: "we assume the internal conformation of side chains to be fixed." This assumption seems too strict because side chains provide hydrogen bonds that form the most important secondary structures.**
>
> We agree with the reviewer that the assumption of fixed side chain conformations is a significant weakness. However, in this paper we focus on what constitutes a major unsolved problem - modeling of constrained loop backbones. As the side-chains are not directly affected by this constraint, their conformations could be addressed using existing approaches, which could be done through a small modification of our architecture. It should be noted that we do not include the side-chain atoms in the calculation of any losses or quality metrics, and the only effect they have is through being present in the input molecular graph. We modify the original sentence to provide some discussion:
> >”For the purpose of this work, we assume the internal conformation of side chains (short molecular chains branching from the $C_\alpha$ atom and specific for each type of amino acid) to be fixed (but note in passing that as side chain conformations are not subject to closure constraints, they can be straightforwardly handled by using existing approaches based on torsional diffusion, and our architecture can be augmented to include such treatment).”
>
> ---
>
> **3. Line 120: "the space of internal protein chain conformations is the hypertorus." Some values of torsion angles lead to clashed atoms involving side chains, hence the realizable space of torsion angles is smaller. See [PNAS Reference](https://www.pnas.org/doi/abs/10.1073/pnas.1014674107).**
>
> We agree with the reviewer that the realizable space of torsion angles is smaller due to steric clashes and other structural constraints. In our work, we delegate the resolution of clashes and formation of feasible conformations to be learned by the diffusion model. The statement about the hypertorus refers specifically to the embedding space of the protein chain, described in terms of torsional angles while keeping other degrees of freedom fixed.
> To clarify this distinction, we have revised the text to state:
> >“ the embedding space of an internal  protein backbone chain is a hypertorus…”
>
>
> ---
>
> **4. Lines 126-127: "Unlike diffusion on Euclidean spaces or smooth manifolds, diffusion on varieties can be challenging in the vicinity of singularities." Though singularities are explicitly mentioned here, Figure 1 and all further constructions discuss smooth manifolds. The word "variety" appears only two more times in the abstract.**
>
> We agree that the discussion of varieties and their associated singularities was insufficiently detailed in the manuscript. To address this, we have expanded the discussion in the reply to the question about *"Lines 285-287"*.
>
> ---
>
> **5. Line 131: "a loop region can move in a finite number of directions to keep the ends fixed." Because atom coordinates are continuous, they can move in infinitely many directions.**
>
> Thank you for pointing this out. You are correct that atom coordinates can move in infinitely many directions. What we intended to convey is that the dimension of the tangent space at a given point on the manifold restricts the possible directions of motion (i.e., not every direction is feasible). To avoid confusion, we have removed this statement from the text.
>
> ---
>
> We will continue in the next message due to character limitations.

---

> ### Author Response · Authors · 2024-11-28
> **Response to Reviewer p9bC (part 2)**
>
> **6. Line 215: "The score model is using a similar architecture based on SE(3)-equivariant convolutional networks over point clouds." If points are not ordered in these clouds, this case is not relevant for protein chains whose atoms are naturally ordered along the backbone. The ordered case is much easier than unordered because a complete SE(3)-invariant has a linear size.**
>
> We apologize for the confusion caused by this phrasing. Our model does not operate on unordered point clouds but rather on a geometric graph that incorporates both the chemical connectivity and spatial proximity of atoms. This graph representation is used to produce torsional scores for the rotatable bonds in the loop.
> The choice of this graph-based representation was motivated by our intent to generalize the applicability of the model beyond protein chains, enabling it to work with arbitrary macrocyclic molecules. We appreciate your observation and have clarified this in the manuscript to avoid further ambiguity.
>
>
> ### Responses to Specific Questions
>
> **Q1. The major weakness is the lack of a rigorous problem statement.**
> We acknowledge this weakness and have clarified and expanded the problem statement in the Introduction to provide a more rigorous definition of the problem we address. Specifically, as detailed in our response to Reviewer 26S9, Question 1.
>
> ---
>
>
> **Q2. Line 17: "we develop a diffusion model that performs a random walk" What is the purpose of this "random walk"?**
>
> We apologize for the confusion caused by this phrasing. Diffusion is the continuous limit of a random walk. We developed a diffusion model on the variety. We adjusted the text accordingly:
> > “Here we develop a diffusion model on the underlying variety by utilizing an appropriate Jacobian, whose loss of rank indicates singularities.”
>
> ---
>
> **Q3. The key concept of a "loop," which appears 80+ times in the paper including the title, is never formally defined. Did the authors mean a geometric embedding of several consecutive amino acid residues or only some atoms (which?) along the backbone without side chains, which are not marked as alpha-helices or beta-strands?**
>
> We apologize for the missing definition. By "loops," we refer to segments of a protein chain that connect regular elements of secondary structures, such as alpha helices and beta sheets, but lack the regularity of hydrogen bonding patterns characteristic of these structures.
> To address this, we have modified the Introduction section to include a formal explanation of the term, as described in our reply to Reviewer 26S9, Question 1. Additionally, we added an illustration in Figure 1 to help visualize and clarify what constitutes a protein loop.
>
> ---
>
> **Q4. Lines 114-115: "the observed variations of bond lengths and angles in experiments are relatively small" How can this claim be justified? For protein chains, bond lengths and angles are not "observed" but are often fixed or refined to "ideal" values. Despite this artificial fixing, the Protein Data Bank has a large percentage of chains whose bond lengths and angles are outside many standard deviations from "ideal" values.**
>
> Thank you for pointing this out. Our statement is based on several references, such as Go & Scheraga, *Macromolecules*, 3(2), 1970, and Dinner, *Journal of Computational Chemistry*, 21, 2000, which note that the variations in bond lengths and angles, while present, are relatively small compared to the degrees of freedom in torsional angles.
> To clarify this point and address the concern, we have included these references in Section 2 of the manuscript to provide proper justification for the statement.
>
> ---
>
> **Q5. Lines 176-177: "(the Implicit Function Theorem) guarantees that six of the variables may be expressed as differentiable functions" This theorem holds only under strict conditions. Why are they satisfied?**
>
> You are correct that the Implicit Function Theorem (IFT) requires specific conditions, particularly the invertibility of the Jacobian. In our formulation of the polynomial equations for loop closure, the chosen six variables are differentiable functions of the others, provided the Jacobian has full rank (6) in a neighborhood of a given point.
> In practice, the rank of the Jacobian matrix depends on the selection of six pivot torsions within the chain. To handle cases where the Jacobian loses rank (e.g., at singularities), we adaptively change the selection of pivot torsions. This approach ensures that the projection process remains feasible and sufficiently robust to deal with singularities. Additionally, our method can handle non-differentiable cases by addressing the rank deficiency directly in the projection process.
> We have expanded the discussion of these considerations in Section 3.2 of the manuscript to provide clarity on how these conditions are managed in our approach.
>
> ---
>
> We will continue in the next message due to character limitations.

---

> ### Author Response · Authors · 2024-11-28
> **Response to Reviewer p9bC (part 3)**
>
> **Q6. Lines 198-202: "We apply R6B6 Cao et al. (2023) algorithm, which is designed to solve closure problem in the chain with fixed ends ... if the closure problem is not solvable by R6B6 which indicating an infeasible movement, the next perturbation will be tried until one solvable case is sampled successfully."**
>
> Can these tries in a continuous space continue infinitely long?
> Theoretically, there is no hard limit to the number of trials in a continuous space. However, the starting structures in our experiments are real 3D conformations with sufficient degrees of freedom, meaning they generally belong to open regions of the conformation space where nearby closed-loop solutions can be found. Singularities, such as those arising from "singularities of projection" or intersections of different components of the conformation space, may cause the Jacobian to lose rank and lead to temporary failures.
> To address such cases, our method adaptively adjusts the perturbations and pivot selection, allowing it to resolve singularities in most scenarios. Empirically, we observe that the closure problem is almost always solved after one trial. Rarely, up to three trials are needed to find a feasible solution.
> To clarify this, we have added the following to Section 3.3:
> > "In our experiments, the closure problem can almost always be solved after one perturbation, with rare failures requiring at most three trials."
>
> **Q7. Lines 285-287: "in the diffusion on toric varieties, the basis vectors for the tangent space at one point on the manifold are dependent on the position of the point, so we need to compute the basis vectors at every step". There is no need to consider any tangent spaces of high-dimensional tori because these manifolds are flat (locally Euclidean in terms of distances and angles), see [Flat Manifold](https://en.wikipedia.org/wiki/Flat_manifold). For a simple example, a 2D torus has two angle coordinates in ranges \([0, 2\pi)\). Why do you need to embed this torus in 3D?**
>
> We agree that tori of any dimension are flat, and that applies to the internal coordinate description of the space of a free chain. In the problems considered here, we need to guarantee that the movement will maintain loop closure, and the main contribution of the paper is to provide a way to accomplish that. Additionally, the reason for considering an \(\mathbb{R}^3\) embedding is to create a framework for uniform coverage in conformation space with respect to the RMSD metric (which can also be used to perform clustering), removing torsional bias (Nilmeier et al., *JCTC*, 2011, 7, 1564). Admittedly, the method presented here does not yet incorporate this element (mainly due to time constraints for the conference), but we felt the introduction of the Plücker coordinate expression (in Equation 2) is adequate to give an intuitive understanding of the need for six degrees of freedom to achieve an open-hand workspace, which one can roughly describe as needing three translational and three rotational degrees of freedom to place the hand at a desired position and orientation in space.
> To clarify these points, we have added more detailed descriptions in Section 3.2 of the manuscript.
>
> ---
>
> **Q8. Line 301: "Each structure was split into two parts: the loop region and the remaining part of the protein." While a loop region wasn't formally defined, a protein structure likely contains many loop regions. Which one is chosen here?**
>
> We agree that defining the term "loop" is important, and we have added a general definition of a protein loop in the Introduction. In principle, our method can be applied to any loop region. However, in the applications presented in the paper, the loops of interest are uniquely defined, with only one such loop present in each system.
>
> Specifically:
> - **MHC Case**: The loop corresponds to the constrained peptide. Since only one peptide is present in each system, it is treated as a single loop.
> - **Nanobody Case**: The loop corresponds to the CDR3 region. Each nanobody has only one CDR3 loop, with its start and end points defined by the widely accepted Chothia numbering scheme. We have added a reference to this scheme in the manuscript for clarity.
>
> We hope this clarification resolves the concern.
>
> ---
>
> We will continue in the next message due to character limitations.

---

> ### Author Response · Authors · 2024-11-28
> **Response to Reviewer p9bC (part 4)**
>
> **Q9. Line 308: "we used the same AF2 model to obtain predicted Local Distance Difference Test (pLDDT) values Mariani et al. (2013)" Do you agree with the conclusion of Mariani et al. (2013) on page 2728: "One disadvantage of the LDDT score is that it does not fulfill the mathematical criteria to be a metric"? If yes, do you know that if a distance fails the triangle axiom with any positive error, then results of clustering can be pre-determined and hence not trustworthy as proved in [this paper](https://ieeexplore.ieee.org/abstract/document/10574843)?**
>
> The reference to Mariani et al. (2013) was misplaced in this context, and we have removed it from the manuscript to avoid any misunderstanding. We used pLDDT (predicted LDDT), which is inspired by LDDT but distinct from the pairwise distance measure discussed by Mariani et al. (2013). pLDDT was used as a conventional AlphaFold2 model confidence estimator. We do not use pLDDT as a metric.
>
> Thank you for bringing this to our attention.
>
> ---
>
> **Q10. Line 310: "We then used these values to rank the structures and determine the confidence of the predictions." How are these predictions justified in the light of the references above?**
>
> As explained in the previous reply, pLDDT is a conventional confidence metric used for quality assessment and ranking of AlphaFold2 models. In this work, we use pLDDT to rank the models generated by our approach, following the standard practice for evaluating AlphaFold2 predictions. This allows us to identify the most reliable conformations from the generated ensemble.
>
> ---
>
> **Q11. Line 320-321: "The testing results showed that closer conformations to the PDB structures exist within the ensemble, demonstrating that the model functions correctly." The loop regions have huge uncertainty in PDB structures because the structure determination methods usually have low resolutions in these regions. Is it possible to justify that the "model functions correctly" by using unreliable "ground truth"?**
>
> We agree with the reviewer about the inherent uncertainty in PDB loop structures due to their flexibility and the limitations of structure determination methods. However, there is currently no better alternative for benchmarking structure prediction methods. The PDB remains the standard reference in the field and has been widely used to validate methods such as AlphaFold and other prediction frameworks.
> Although the loop regions are inherently flexible, the PDB provides a reasonable reference point for comparison, and improvements in RMSD and other metrics against PDB structures are indicative of the model’s ability to generate biologically plausible conformations.
>
> ---
>
> **Q12. How large are the databases discussed in Sections 4.1 and 4.2?**
>
> We appreciate the question and recognize the need to provide this information explicitly in the main text.
> - **MHC Dataset**: There are 789 structures in total. We split the dataset into training, validation, and test sets, using 636 structures for training and evenly dividing the remainder between validation and testing.
> - **Nanobody Dataset**: There are 505 structures in total. Similarly, we trained the model with 403 structures and evenly split the remaining data for validation and testing.
>
> We have added these dataset size details to the main text to ensure clarity.
>
> ---
>
> **Q13. Lines 13-14: "Toric varieties. These are real algebraic varieties, formulated as the zero sets of polynomial equations constraining the rotor angles in a linkage or macromolecular chain." The abstract promised to describe toric varieties of macromolecular chains by polynomial equations. In what line numbers did these equations appear if the word "polynomial" was found twice in the abstract and never again?**
>
> We acknowledge the oversight and apologize for the confusion. To address this, we have added a brief description of the derivation of the polynomial equations that define the toric varieties in Section 3.1. These equations explicitly describe the geometrical constraints underlying loop closure. This update clarifies the connection between the toric variety representation and the underlying algebraic formulation, as outlined in the response to Question 7.

---

> > ### Comment · Reviewer_p9bC · 2024-11-28
> > **Thank you for the reply**
> >
> > I sincerely thank the authors for their detailed and thoughtful responses. Taking in account the revision, I increased the scores for the presentation and contribution, and also the overall score to 5. The remaining concerns are (1) the heuristic definition of a loop region as a complement to alpha-helices and beta-strands, which are also heuristically defined by using past algorithms based on manually chosen thresholds, (2) using the atomic coordinates in the PDB as a ground truth, where resolutions are rarely below 1 Angstrom, (3) the black-box nature of the model, which trains many hidden parameters and uses computational non-experimental AlphaFold2 outputs as inputs. The authors are encouraged to think about a rigorous explainable approach to the loop problem.

---

### Official Review · Reviewer_26S9 · 2024-11-03

**Soundness:** 3
**Presentation:** 1
**Contribution:** 2
**Rating:** 3
**Confidence:** 2

**Summary:**

The paper introduces a diffusion model that allows to explore toric algebraic varieties. The authors demonstrate an improvement upon the state-of-the-art AlphaFold approach on two protein structure prediction data sets.

**Strengths:**

The structure of the paper is good, and the choice of experiments (data sets, baseline method) seem to be appropriate.

**Weaknesses:**

The paper is extremely difficult to read, I think it is inaccessible to the majority of the wide ICLR audience. This is the main reason behind my rejection, I am willing to raise my score if the main goal and contributions of the paper are much more clearly described.

For instance, after reading the Abstract and Introduction, it was still not clear to me what the problem is one is aiming to solve. Formulations such as "molecular modeling", "molecular structure generation" and "protein prediction" are used, but the abstract also mentions kinematic linkages in robotics, so what is the (precise) formulation of the general problem? In addition, so much jargon is used, such as "conformer sampling" (already in the very first paragraph, without any further context or explanation), "number of flexible torsions in the loop", "energetics", "rotors", ... Related work is using too much jargon too, and is almost unreadable for general audience.

See Questions below for more examples, that could help to make the paper clearer and of interest to a wider community. For these reasons, it was impossible for me to properly assess the quality of this manuscript.

**Questions:**

(Q1) What is precisely the problem you are aiming to solve? As I mention above, you use "molecular modeling", "molecular structure generation" and "protein prediction", but at a later point in the Introduction you also talk about learning a constrained high dimensional manifold. If the latter also applies, should the Related work not also discuss the manifold learning literature?

(Q2) What is the (idea behind) diffusion model and why is it useful for this problem? In Introduction, you explain this with "Since conformer sampling can be treated as a generative problem, many diffusion models have been proposed to produce molecular conformers ...", but this is not very useful if one does not yet understand the problem (Q1), nor is familiar with conformers.

(Q3) The result from Angeles (2014) that guarantees that six of the torsions can be determined once other n-6 torsions are assigned is only mentioned as such in the Introduction, but then heavily used throughout the paper. Can you at least provide some intuition behind this results, where is the number 6 coming from? Also, since this is a book of almost 600 pages, it would be useful to point to a specific result (theorem or at least a section).

(Q4) At the end of the Related work section, you mention that AF can be run several times starting from different random seeds. Until this moment, the reader does not even know that the method is not deterministic. Since you use this approach as the baseline you compare against, can you describe it briefly, how does it work? Which group of methods does it belong to, from the ones you mention early in this section? What are the crucial differences between AF and your proposed approach?

(Q5) The score model is using a similar architecture based on e3nn and tensor field network that have not been peer reviewed and have respectively around 150 and 1000 citations, why not PointNet that has over 15000 citations and is commonly used as a benchmark deep learning architecture for point clouds?

(Q6) What is Å?

(Q7) Why is AF3 used in Section 4.1, but not also in Section 4.2?

(Q8) In the Conclusion (and all of a sudden), you write that "Previous studies on protein structure prediction have maintained that relationships between residues at long distances cannot be learned using a diffusion model." What is this based on, what is the underlying logic and why did you decide for a diffusion model despite these insights?

Typos and minor comments:
- line 088: conformerss
- line 089: varierty
- line 298: Appendix Table 3 -> Appendix A Table 1
- References: The references should be properly formatted in the main text (so that they appear in brackets whenever appropriate, choose carefully between \citet and \citep). Also, in the References section, improve the formatting, e.g., capitalization of journals (Advances in neural information processing systems).

**Details Of Ethics Concerns:**

The Acknowledgment section explicitly mentions some NIH and NSF grants, centres and facilities, not sure if this violates the submission guidelines / anonymity policy.

---

> ### Author Response · Authors · 2024-11-28
> **Response to Reviewer 26S9 (part 1)**
>
> **General Comment**
>
> We thank the reviewer for the careful consideration of our manuscript. After familiarizing ourselves with the reviewer’s comments, we agree that the text in its original form was not sufficiently clear or accessible to the general audience and may have excessively relied on subfield-specific lingo and assumptions of common knowledge. We made our best effort to improve the clarity of the text and address the points raised.
>
> ---
>
> ### Responses to Specific Questions
>
> **Q1. What is precisely the problem you are aiming to solve?**
>
> Our goal is to address the challenge of protein loop structure prediction, a critical subproblem in protein structure prediction. Protein loops are non-regular segments connecting regular structural elements (e.g., alpha helices and beta sheets) and are often exposed to the solvent, making them difficult to characterize experimentally and computationally.
> We agree that our initial manuscript lacked clarity in defining this problem. To address this, we have revised the first three paragraphs of the introduction to better highlight the importance of protein loop structures, their modeling challenges, and how our work contributes to solving this issue.
>
> **Q2. What is the (idea behind) diffusion model and why is it useful for this problem?**
>
> We agree with the reviewer that it was left unclear in the initial version of the paper. We have updated the Introduction section to explain the motivation and significance of constructing a new diffusion-based model.
>
> **Q3. The result from Angeles (2014) that guarantees that six of the torsions can be determined once other n-6 torsions are assigned is only mentioned as such in the Introduction, but then heavily used throughout the paper. Can you at least provide some intuition behind this result? Where is the number 6 coming from?**
>
> A derivation of the closure equations in terms of six variables is found, e.g., eq. 9.27, p.389 in Angeles. We have updated the discussion in Section 3.2 to provide a clearer intuition behind this result:
> > “Intuitively, the linkage needs 6 DoF to maintain closure, since given the location of one end, placing the other at the correct position and orientation requires, roughly speaking, 3 translational and 3 rotational degrees of freedom.“
>
> **Q4. At the end of the Related Work section, you mention that AF can be run several times starting from different random seeds. Until this moment, the reader does not even know that the method is not deterministic. Since you use this approach as the baseline you compare against, can you describe it briefly? How does it work? Which group of methods does it belong to, from the ones you mention early in this section? What are the crucial differences between AF and your proposed approach?**
>
> We expand and clarify the description of the AF2 model and its relevance as a baseline for comparison in Section 2: Background and Related Work. We also now cover the process of generating diverse results with AF2 in the third paragraph of the Experiments section.
>
> **Q5. The score model is using a similar architecture based on e3nn and tensor field networks that have not been peer-reviewed and have respectively around 150 and 1000 citations. Why not PointNet, which has over 15,000 citations and is commonly used as a benchmark deep learning architecture for point clouds?**
>
> Our architecture is similar to SE(3)-equivariant Tensor Field Networks, chosen for their robust handling of geometric symmetries. The primary reason for this choice is to leverage the SE(3) symmetry inherent to protein structure prediction problems. It helps avoid issues related to overall protein orientation and the absolute positions and rotations of individual residues.
> We acknowledge that alternative architectures, such as PointNet, could also be employed. This possibility has been noted in Section 3.4 for potential future exploration.
>
> **Q6. What is Å?**
>
> We apologize for the lack of clarity.
> Å, or Ångstrom, is a unit of length often used in the field of structural biology and equal to \(10^{-8}\) cm. We add a short explanation at the first mention of this unit in the Experiments section.
>
> ---
>
> We will continue in the next message due to character limitations.

---

> ### Author Response · Authors · 2024-11-28
> **Response to Reviewer 26S9 (part 2)**
>
> **Q7. Why is AF3 used in Section 4.1, but not also in Section 4.2?**
>
> We thank the reviewer for pointing out this discrepancy. We have updated Section 4.2 to include the AF3 results.
>
> **Q8. In the Conclusion (and all of a sudden), you write that "Previous studies on protein structure prediction have maintained that relationships between residues at long distances cannot be learned using a diffusion model." What is this based on, what is the underlying logic, and why did you decide on a diffusion model despite these insights?**
>
> We thank the reviewer for pointing out this issue. In our work, we focus on modeling close-range interactions within protein loops, which aligns well with the capabilities of diffusion models. The referenced statement in the Conclusion was misplaced and not relevant to the context of our work. We have removed this sentence from the manuscript to avoid confusion.
>
> ---
>
> We hope the responses address the reviewer’s concerns, and we will update the manuscript accordingly to reflect these clarifications and corrections. Thank you for your valuable feedback.

---

### Official Review · Reviewer_RD41 · 2024-11-04

**Soundness:** 3
**Presentation:** 2
**Contribution:** 2
**Rating:** 5
**Confidence:** 3

**Summary:**

This paper proposes a diffusion model on the conformation space of loop regions in proteins, using tools from inverse kinematics in robotics. In particular, for a given protein loop region, they solve for the basis vectors of the manifold of torsion angles which maintain the position of the loop ends, using singular value decomposition. They then add noise to the given torsion angle along this manifold, ensuring loop closure with the R6B6 algorithm. They then seem to learn the score function w.r.t. this diffusion process. Experimental results indicate improved performance compared to AlphaFold 2 and 3. Overall the paper was reasonably well-written and proposes an interesting approach. However, I am unsure about the significance and motivation of the approach. Additionally, I felt that more elaboration is needed on some aspects, e.g. the score-matching algorithm and the method R6B6.

**Strengths:**

1. Presents a novel approach, to the best of my knowledge.
2. Make an interesting connection between molecules and robotics.
3. Takes advantage of AlphaFolds predictions as inputs, making much more data available.

**Weaknesses:**

1. Can be cumbersome to do an SVD of the Jacobian matrix at every step.
2. Not much elaboration on the score-matching part of the algorithm.
3. Uses an external method, R6B6, to ensure loop closures, but does not elaborate much on it.

**Questions:**

1. What is the cost of computing the SVD of the Jacobian at every step compared to classical geometric methods?
2. How much does the change in loop conformation affect the conformation of the rest of the protein?
3. Could you please elaborate more on the use of R6B6

**Details Of Ethics Concerns:**

The authors seemed to forget to remove the acknowledgments (although I did not notice until the end of the review).

---

> ### Author Response · Authors · 2024-11-28
> **Response to Reviewer RD41**
>
> **General Comment**
>
> Thank you for your detailed review and insightful feedback. We appreciate your recognition of the novelty of our approach and its connection to robotics-inspired methods. Your comments have helped us identify areas where further elaboration and clarification were needed. Below, we summarize the key revisions made to the manuscript and provide detailed responses to your questions.
>
> ---
>
> **Key Revisions Made**
> 1. **Clarified Score-Matching Algorithm**:
> - Expanded Section 3.3 to describe the score-matching loss function and its role in training the diffusion model.
> - Revised Section 3.4 to clearly explain the architectural details and highlight the novel elements of our approach.
> 2. **Described R6B6 in More Detail**:
> - Added a concise explanation in Section 3.1, describing R6B6’s role in ensuring loop closure and how it operates within our diffusion framework.
> 3. **Efficiency of SVD Computation**:
> - Added a discussion in Section 3.3 to demonstrate the negligible computational cost of SVD compared to classical geometric methods like TLC.
> 4. **Clarified Impact on Non-Loop Regions**:
> - Updated Section 3.4 to explicitly state that non-loop regions are fixed and provide spatial constraints to guide loop movements.
>
> ---
>
> **Answers to Specific Questions**
>
> **1. What is the cost of computing the SVD of the Jacobian at every step compared to classical geometric methods?**
>
> The computational cost of SVD for our \(6 \times N\) Jacobian matrix is \(O(6N \, \min(6, N))\), which is linear in \(N\). Since \(N\) is at most 34 in our use case, this results in a computation time on the order of \(10^{-5}\) seconds per step. Given that one diffusion step requires less than 0.1 seconds overall, the cost of SVD is negligible compared to the benefits it provides in efficiently sampling the manifold.
> In contrast, classical geometric methods like TLC require generating thousands of conformations (e.g., >30,000 for loops with 14 residues), resulting in an overall cost that is orders of magnitude higher due to additional clustering and energy-based scoring steps. As noted above, our approach achieves similar or better results with only 20 conformations, making it significantly more efficient.
> We have clarified this in Section 3.3 of the manuscript.
>
> ---
>
> **2. How much does the change in loop conformation affect the conformation of the rest of the protein?**
>
> In the current implementation, the rest of the protein remains fixed throughout the process, and only the loop region is allowed to move. While the rest of the protein may constrain the loop’s movements, changes in the loop conformation do not affect the non-loop regions. We have clarified this in Section 3.4 by adding:
> > “Non-loop regions are fixed in the input graph, serving as spatial constraints to guide feasible loop movements while ensuring closure.”
>
> ---
>
> **3. Could you please elaborate more on the use of R6B6?**
>
> We have added a detailed description of R6B6 in Section 3.1, emphasizing its role in maintaining loop closure by solving for the 6 constrained degrees of freedom while perturbing the remaining \(n - 6\) torsions. The revised section also includes:
> > "R6B6 (from “6 Rotors/6 Bars”) is a robust algorithm designed to handle loop closure and conformational sampling problems in chains with fixed ends. It uses a system of polynomial equations to solve for the constrained torsions, ensuring the chain remains closed while allowing flexible perturbation of the remaining torsions. In a chain with n flexible torsions, we can select n − 6 torsions to perturb, and R6B6 can be used to solve for the remaining 6 torsions to maintain two ends of the chain fixed with respect to the rest of the protein structure."
>
> ---
>
> We hope these clarifications and revisions address the concerns raised and improve the overall clarity and impact of the manuscript. Thank you again for your valuable feedback.

---

### Official Review · Reviewer_q7YB · 2024-11-09

**Soundness:** 3
**Presentation:** 2
**Contribution:** 3
**Rating:** 5
**Confidence:** 2

**Summary:**

The paper focuses on the problem that the loop regions usually have lower accuracy compared to other structures in the protein prediction tasks. The paper proposes a diffusion model to generate loop conformations in toric varierty space. The performance of the method was tested using the MHC dataset and the nanobody dataset. It achieves improvement upon AlphaFold 2 by 15% and comparable performance to AlphaFold 3.

**Strengths:**

- The paper focuses on the loop regions in the protein, which is a useful topic to improve the protein prediction results.
- The paper proposes a novel diffusion process on toric varieties to generate conformations for protein loop regions with constrained ends.

**Weaknesses:**

- Although the paper selects "No Acknowledgement Section", there is an acknowledgement section in the paper...
- Besides, I just have some additional questions listed in the "question" section.

**Questions:**

- How long does it take to run each inference step?
- Does the diffusion model also takes non-loop regions as input information?
- In Figure 2, the paper compares results with AF2(1) and AF2(20) and AF3. Was there a specific reason to not compare with AF3 in the similar way as AF2 (i.e., also compare the best results of 20 random seeds)?
- The method first uses AF2 to predict the protein structures and uses that as inputs. The diffusion model then improve the protein prediction results by generated conformations for the loop regions. It achieve improvement upon AlphaFold 2 by more than 15% and comparable performance to AlphaFold 3.  - If the input protein structures are generated by other methods (such as AF3) instead, will the proposed method also give the similar amount of improvement upon the initial structures?

---

> ### Author Response · Authors · 2024-11-20
> **Response to Reviewer q7YB**
>
> **General Comment**
> We would like to sincerely thank the reviewer for their thoughtful feedback and constructive suggestions. We appreciate the recognition of our contributions and the insightful questions raised. We apologize for the oversight regarding the Acknowledgements section, which was mistakenly included despite our selection of "No Acknowledgement Section." We will correct this in the revised manuscript.
>
> ---
>
> ### Responses to Specific Questions
>
> **1. How long does it take to run each inference step?**
> The inference process generates one conformation in approximately 1 second, which includes 20 diffusion steps. This indicates that each step takes less than 0.05 seconds on average. We will include this timing detail in the revised manuscript.
>
> **2. Does the diffusion model also take non-loop regions as input information?**
> Yes, non-loop regions are included in the input graph, but they remain fixed throughout the process. These regions provide spatial constraints, ensuring that the loop movements maintain closure and feasibility. We will clarify this in the relevant section (Section 3.4) by adding the following text: *"Non-loop regions are fixed in the input graph, serving as spatial constraints to guide feasible loop movements while ensuring closure."*
>
> **3. In Figure 2, the paper compares results with AF2(1), AF2(20), and AF3. Was there a specific reason not to compare AF3 in a similar manner to AF2 (i.e., also compare the best results of 20 random seeds)?**
> The comparison with AF3 under the same conditions as AF2 (20 random seeds) was not feasible due to server usage constraints. Specifically, the AlphaFold 3 (AF3) server limits users to 20 job submissions per day, making it impractical to run 20 random seeds for nearly 80 structures within the review timeline. This limitation was the primary reason for not including this comparison. We will add this explanation to the manuscript to clarify our methodology.
>
> **4. If the input protein structures are generated by other methods (such as AF3) instead of AF2, will the proposed method still provide a similar level of improvement upon the initial structures?**
> Due to the usage terms of AF3, we are unable to use its structures for training or refinement with our diffusion model. However, we provide comparisons between our method and AF3 predictions to evaluate performance. The results indicate that our method performs comparably to AF3 on both datasets. If future usage terms allow, this is a potential avenue for further investigation.
>
> ---
> We hope the above responses address the reviewer’s concerns and will update the manuscript accordingly to reflect these clarifications and corrections. Thank you for your valuable feedback.

---

### Meta-Review · Area_Chair_Pfg7 · 2024-12-19

**Metareview:**

This paper proposes a diffusion generative model for looped structures by leveraging toric varieties formed from systems of polynomial equations. The applications to two protein structure prediction problems are considered. Both the reviewers and me found the approach interesting, but there are also major concerns about significance, innovation, and presentation. Recognizing the potential of the idea, I recommend the authors take reviewers' comments into consideration and submit again.

**Additional Comments On Reviewer Discussion:**

Both the reviewers and me found the approach interesting, but there are also major concerns about significance, innovation, and presentation.

---

### Decision · Program_Chairs · 2025-01-22

Reject